# Learning Efficient Representations of Neutrino Telescope Events

## Abstract

Neutrino telescopes detect rare interactions of particles produced in some of the most extreme environments in the Universe. This is accomplished by instrumenting a cubic-kilometer volume of naturally occurring transparent medium with light sensors. Given their substantial size and the high frequency of background interactions, these telescopes amass an enormous quantity of large variance, high-dimensional data. These attributes create substantial challenges for analyzing and reconstructing interactions, particularly when utilizing machine learning (ML) techniques. In this paper, we present a novel approach, called `om2vec`, that employs transformer-based variational autoencoders to efficiently represent neutrino telescope events by learning compact and descriptive latent representations. We demonstrate that these latent representations offer enhanced flexibility and improved computational efficiency, thereby facilitating downstream tasks in data analysis.

## 1 Introduction

Neutrino telescopes search for rare interactions caused by neutrinos, an elusive particle central to many open questions in the Standard Model of particle physics, such as the origin of their masses and the matter anti-matter asymmetry of the universe. The leading detector in operation today, the IceCube Neutrino Observatory, comprises 5,160 optical modules (OMs) arranged on strings deep within the clear ice of the Antarctic glacier (see Aartsen et al. (2017)). Each OM is designed to precisely record the arrival times of photons, forming a photon arrival time distribution (PATD). The data gathered by IceCube are archived as "events," which are triggered when a specific number of coincident light detections occur between OMs in neighboring strings, signaling the presence of a charged particle. Figure 1 is an artistic representation of the photons produced in a neutrino interaction and their subsequent detection in the OMs. A neutrino telescope typically classifies events into categories based on the morphology of the light detected: cascade-like events, identifiable by their approximately spherical spatial distribution; and track-like events, characterized by their elongated, linear spatial morphology. In IceCube, events are recorded at an approximate rate of 3000 per second, constituting a large data rate. Moreover, comparable or even larger optical neutrino telescopes are either under construction or planned, such as KM3NeT (Aiello et al., 2019), Baikal-GVD (Avrorin et al., 2018), P-ONE (Agostini et al., 2020), IceCube-Gen2 (Aartsen et al., 2021), TRIDENT (Ye et al., 2022), and HUNT (Huang et al., 2023), signaling a need to develop methods for efficiently handling neutrino telescope data.

The accurate reconstruction and prediction of the interacting particle's physical properties—including its energy, direction, and type— from the PATDs recorded by each OM, is essential for informing and facilitating downstream physics analyses. A significant challenge in this task is the incorporation of the full-timing information provided by the PATDs of the OMs, which are characterized by high dimensionality and variable lengths. Interaction events may last several $\mu s$, yet ns-scale timing resolution is needed on the PATDs for most downstream physics analyses. Furthermore, as illustrated by OM#1 in Figure 1, some OMs near the interaction point can detect up to tens of thousands of photons in the highest-energy and brightest events. On the other hand, many further OMs record only a few photons, resulting in highly irregular events; illustrated in the OM#2 distribution on the same figure. In this paper, we introduce a transformer-based variational autoencoder (VAE), termed `om2vec`, which encodes PATDs into a fixed-size compact latent representation. `om2vec` is trained to reconstruct the input PATD as readout by OMs, using only the information

from its encoded latent space. In this sense, it learns an efficient and more flexible representation of the data recorded in each OM. Our approach provides several significant advantages over existing methods, including improved information retention, greater reliability, faster processing speed, and reduced hyperparameter dependence. We also demonstrate that utilizing transformers for this particular problem is particularly effective, by comparing it to a baseline fully-connected network. We argue that these advantages make this the first "one-size-fits-all" neutrino event representation learner, making it suitable for deployment at the earliest stages of experimental data collection. This will serve to circumvent the challenges inherent in raw neutrino telescope data, and facilitate the rapid adoption of more sophisticated ML techniques from the image-processing community into the neutrino physics field. Source code, datasets, and pre-trained checkpoints can be found on GitHub.

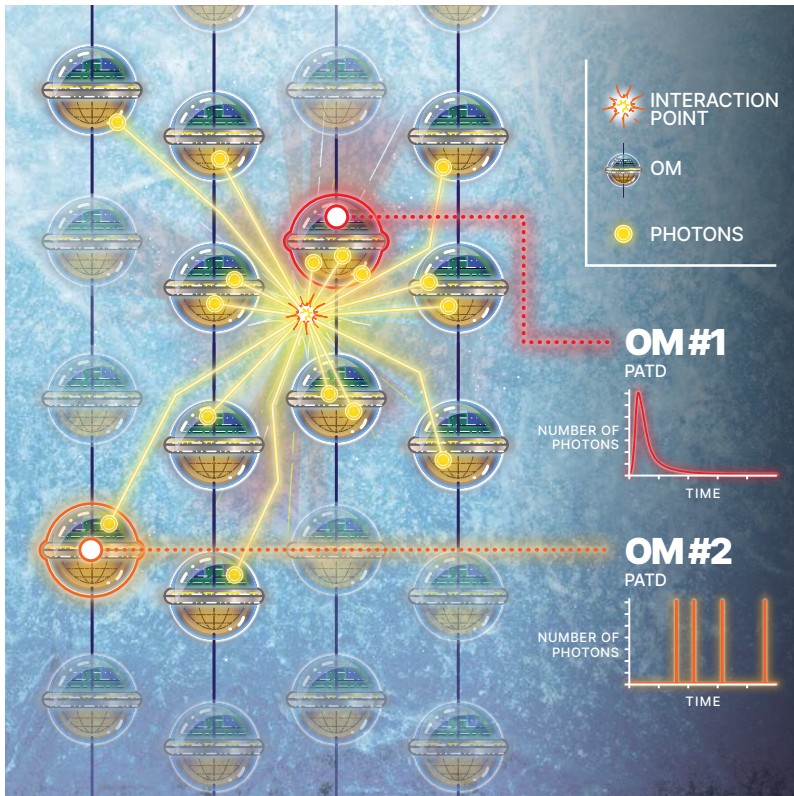

Figure 1: An artistic rendition of a cascade-like event, showcasing how photon arrival time distributions (PATDs) are recorded in neutrino telescopes. A neutrino interaction producing photons occurs in the detector medium, where it is surrounded by photon-detecting optical modules (OMs). The photon arrival times are recorded and counted, as shown in the histograms in the above figure. The amount of photons a particular OM sees is highly variable and generally depends on its proximity to the interaction point. The main goal of om2vec is to convert the PATD on each OM in the event into a fixed-size latent representation.

## 1.1 RELATED WORKS

Machine learning techniques have successfully been applied to neutrino reconstruction tasks in recent years, notably in neutrino telescopes such as IceCube (Choma et al., 2018; Abbasi et al., 2021; Huennefeld et al., 2021; Abbasi et al., 2022; Yu et al., 2023; Eller, 2023; Jin et al., 2024; Zhu et al., 2024) and KM3NeT (De Sio, 2019; Aiello et al., 2020; Reck et al., 2021; Guderian, 2022; Mauro & de Wasseige, 2023). Previous studies have employed various approaches to address the challenges of working with neutrino telescope data. One approach uses summary statistics (Abbasi et al., 2021), utilizing key variables that summarize the PATD of a given OM. This method yields greater efficiency and flexibility, as each OM has a fixed-size description of the timing distribution, allowing events to be formatted as 2D images with multiple features per pixel. However, the

summary statistics lose a substantial amount of information present in the original PATD. An alternative method involves parameterizing the PATD by fitting an asymmetric Gaussian mixture model (AGMM). In this study, we implemented a basic AGMM to serve as a comparison, drawing inspiration from Huennefeld et al. (2021). While this retains significantly more information than summary statistics, the optimization process can be slow and prone to failure, with a strong dependence on hyperparameters, e.g., the number of Gaussian components. Our transformer-based VAE approach builds upon these previous methods while addressing their limitations.

## 2 ARCHITECTURE

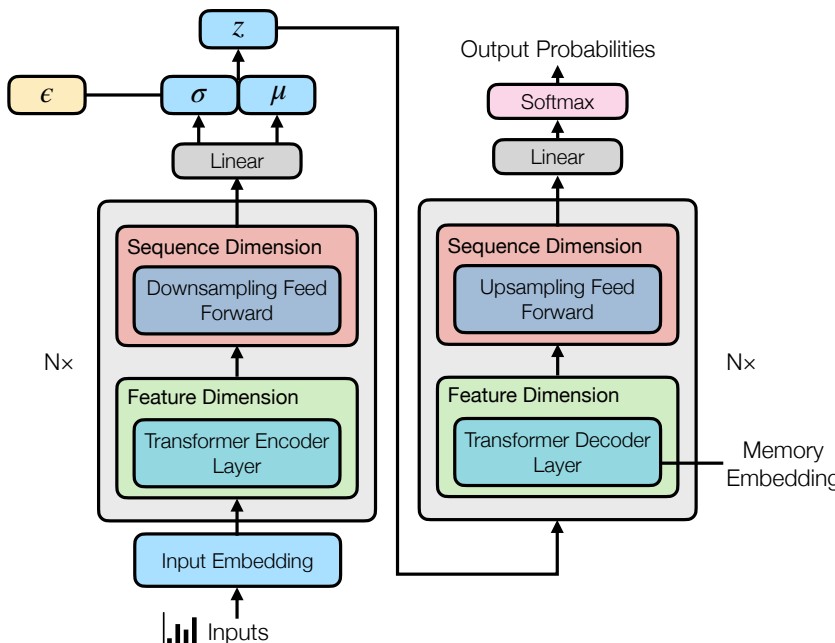

Figure 2: Model architecture of `om2vec`. Input embeddings are generated from the binned PATDs. Each encoder (decoder) block operates on both the feature and sequence dimensions, where feed-forward layers are used to downsample (upsample) the length of the distributions. In the decoder, a memory embedding is learned to keep the decoder independent of the encoder.

Since their inception, VAEs have been extensively utilized for learning effective data representations and generative modeling (Kingma & Welling, 2022). More recently, transformers have demonstrated exceptional performance in handling sequential data (Vaswani et al., 2017). Given the time series nature of our dataset, we incorporate transformer layers to enhance feature learning capabilities. Our model employs the standard encoder-decoder structure, as illustrated in Figure 2. We assume a Gaussian prior for the latent space and train the encoder to learn the parameters $\mu$ and $\sigma$, with some added random noise $\epsilon$. The re-parameterization trick is then utilized to construct the latent representation $z$ while maintaining proper gradient flow to these parameters. A key property of VAEs over regular autoencoders is their continuous latent space, meaning that similar representations within this space correspond to similar reconstructed PATDs.

The input data PATDs are formatted as one-dimensional time series sequences of equal length, with each element corresponding to a specific time in sequence. Each element features a single attribute: the number of photon hits occurring during that particular time window. We first expand the feature space with an input embedding layer. The encoder and decoder are constructed by stacking blocks. Importantly, we employ a hybrid approach that operates across both the feature and sequence dimensions. Transformer layers process the feature dimension, followed by a feed-forward network that down-samples or up-samples the sequence length in the encoder and decoder, respectively. Linear layers are then used to flatten the feature dimension into latent representations, as well as to generate output logits after the decoder. We use a simple vector of learnable parameters, the "memory em-

bedding," which acts as the memory input for the transformer decoder layers. After the final linear layer, the outputs are fed through the softmax function to obtain a properly normalized probability density. We interpret the final softmax activation's output as the probability of detecting a photon hit in each timing bin.

## 3 DATASETS

Simulations play a key role in neutrino physics experiments, facilitating preliminary testing that informs experiment development and subsequent physics analyses. As such, the field has developed sophisticated simulation tools capable of generating events for incorporation into our training datasets. In this paper, we present results trained and tested on simulated events, but emphasize that the methodology can be readily applied to real detector data. We employ the open-source simulation toolkit `Prometheus` (Lazar et al., 2024) to emulate an IceCube-like array of OMs within Antarctic ice. We generate four datasets of events based on the different neutrino interaction types, corresponding to the distinct flavors of the neutrino, $\nu$. Specifically, these interactions are the charged-current electron neutrino ($\nu_e$ CC), muon neutrino ($\nu_\mu$ CC), and tau neutrino ($\nu_\tau$ CC) interactions, alongside the neutral-current ($\nu$ NC) interactions. Notably, the $\nu_e$ CC, $\nu_\tau$ CC, and $\nu$ NC interactions produce predominantly spherical cascade-like events, while the $\nu_\mu$ CC interactions result in long track-like events. Additionally, as further detailed in Section 5, the PATDs generated by $\nu_\tau$ CC interactions are of particular interest to physicists due to their potential to exhibit a smoking-gun "double-bang" signature of astrophysical neutrinos.

Events are simulated, and all OMs detecting at least one photon hit are included in the overall training dataset. We extract five million PATDs from both $\nu_e$ CC and $\nu$ NC interactions, as their signals are quite similar. Additionally, 7.5 million PATDs are sourced from $\nu_\mu$ CC events, and 3.25 million PATDs from $\nu_\tau$ CC interactions, totaling approximately 20 million PATDs for the training dataset. An additional 4.5 million distributions are reserved for the test dataset, mixed roughly equally from each interaction type. The majority of OMs registering light detect only a few photon hits. For an illustration of this distribution, see the bottom panel in Figure 3. Notably, photon yields from interactions increase significantly at higher energies. The simulated events span a wide range of energies to encompass all types of PATDs that may occur in neutrino telescopes.

As previously mentioned, each PATD in the datasets is pre-processed into a fixed-length vector by binning photon hits over time. A total of 6400 bins, each 1 ns wide, represent the first 6.4 $\mu s$ of hits—a readout time frame consistent with IceCube's OMs (Aartsen et al., 2017). The choice of binning and time window can be adapted to suit the specifications of other neutrino telescopes. Any remaining hits are accumulated in a final overflow bin. The hit counts in each bin are then log-normalized, as they can be extremely large for high-energy neutrino interactions.

## 4 PROPOSED METHODS

In this section, we provide a more detailed definition of the objective function for this problem. As previously discussed, the conceptual goal of `om2vec` is to learn how to encode a PATD into a representation vector and subsequently decode this representation back into its corresponding probability density function. Mathematically, the reconstruction loss is defined as

$$\mathcal{L}_{reco}(\theta) = -\log \mathcal{P}(\mathbf{n}|\theta) = \sum_{i=1}^{N_{\text{bins}}} \left[ -n_i \log(\lambda_i(\theta)) + \lambda_i(\theta) \right] \tag{1}$$

where:

$$n_i = \text{observed count in bin } i \text{ (true PDF)}$$
$$\lambda_i(\theta) = \text{expected count in bin } i \text{ (predicted PDF)}$$
$$N_{\text{bins}} = \text{total number of bins, or input length}$$
$$\theta = \text{model parameters}$$

Given the predicted PDF from the network (after softmax), and the normalized true PDF from the input PATD, this loss function takes the negative logarithm of the summed Poisson likelihoods,

representing the probability of registering the observed number of hits in each bin, based on the predicted hit counts.

The total training loss combines reconstruction loss and a KL divergence regularization loss on the latent space. This regularization term is needed to ensure that the learned latent space aligns with the prior normal distribution. It is also scaled by a $\beta$ factor that follows a cyclic cosine function during training, peaking at a hyperparameter value set at $10^{-5}$. A batch size of 1024 is used during training.

## 5 RESULTS

In this section, we present the performance and efficiency results of `om2vec` across several benchmarks. In several cases, we also compare to the more traditional AGMM method for encoding PATDs. The AGMM operates by fitting multiple asymmetric Gaussians defined by the equation

$$AG(t \mid \mu, \sigma, r) = \frac{2}{\sqrt{2\pi} * \sigma(r+1)} * \begin{cases} \exp(-\frac{(t-\mu)^2}{2\sigma^2}), & x \leq \mu \\ \exp(-\frac{(t-\mu)^2}{2(r\sigma)^2}), & \text{otherwise} \end{cases}$$

where $\mu$ and $\sigma$ are the mean and standard deviation and $r$ is the asymmetry parameter, as introduced in (Huennefeld et al., 2021). Each asymmetric Gaussian is assigned a weighting factor, $w$, and then summed. Consequently, each component of the mixture model introduces four parameters: $\mu$, $\sigma$, $r$, and $w$. The mixture model is initially fit using $k$-means clustering to obtain the starting parameters, followed by solving a bounded-constrained optimization problem that minimizes the negative log-likelihood. However, the fit is only performed when the number of photon hits (the length of $x$) is greater than the number of Gaussian components. Otherwise, the true photon hit times are stored.

### 5.1 TIMING DISTRIBUTION REPRODUCIBILITY

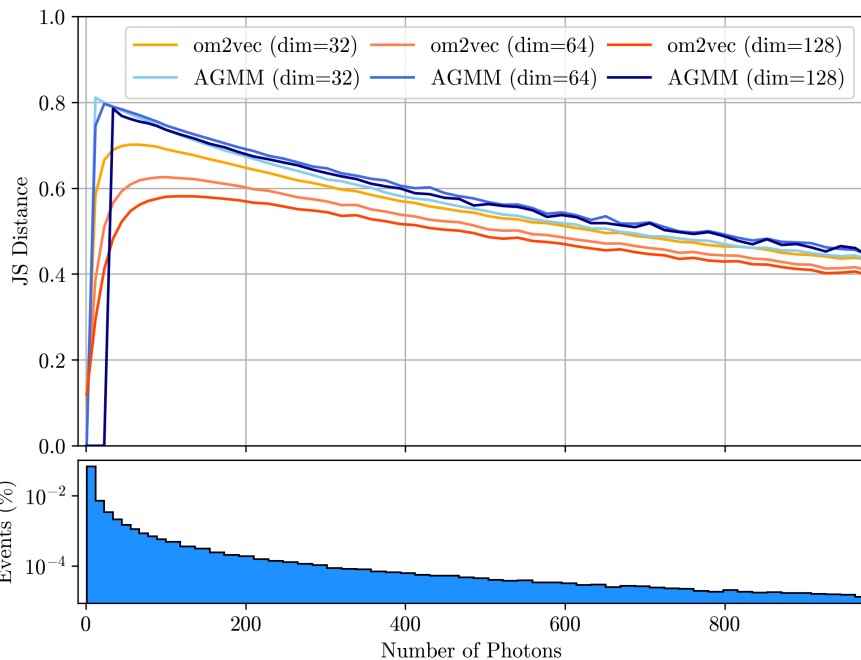

Figure 3: JS distance between the true input PATD and the reconstructed PATD across different methods, plotted as a function of the number of detected photons in the input PATD. Greater values of the JS distance implies a worse fit. In the AGMM methods, the sudden jump is caused by the fit only being performed when the number of photon hits is greater than the number of Gaussian components. The lower panel illustrates the percentage of PATDs in each number of photons bin, relative to the total dataset.

We now evaluate the ability of om2vec to retain information in its latent representation by measuring how accurately it can reconstruct the input PATD. We train three separate models, each with a different latent dimension size (32, 64, and 128). For comparison, we also present results from our AGMM implementation, using the same total number of parameters (8, 16, and 32 components, with 4 parameters per component).

To assess the reconstruction capability, we employ the Jensen-Shannon (JS) distance (Lin, 1991; Nielsen, 2019), which is derived from the KL divergence and is defined as follows:

$$D_{JS}(P \mid Q) = \sqrt{\frac{1}{2}D_{KL}(P \mid \frac{P+Q}{2}) + \frac{1}{2}D_{KL}(Q \mid \frac{P+Q}{2})},$$

given the distributions $P$ and $Q$, and where $D_{KL}$ is the KL divergence. The JS distance provides a similarity score between 0 and 1, where 0 indicates identical distributions and 1 indicates maximum difference. In Figure 3, we present the median JS distance curves for both om2vec and AGMM approaches, across the three different latent dimension sizes. We plot the JS distance as a function of the number of photons recorded in the PATD, scaling up to a thousand photons. While some PATDs observe significantly more photons from the highest-energy events, the statistics are sparse (i.e., there are few PATDs) at those higher photon counts.

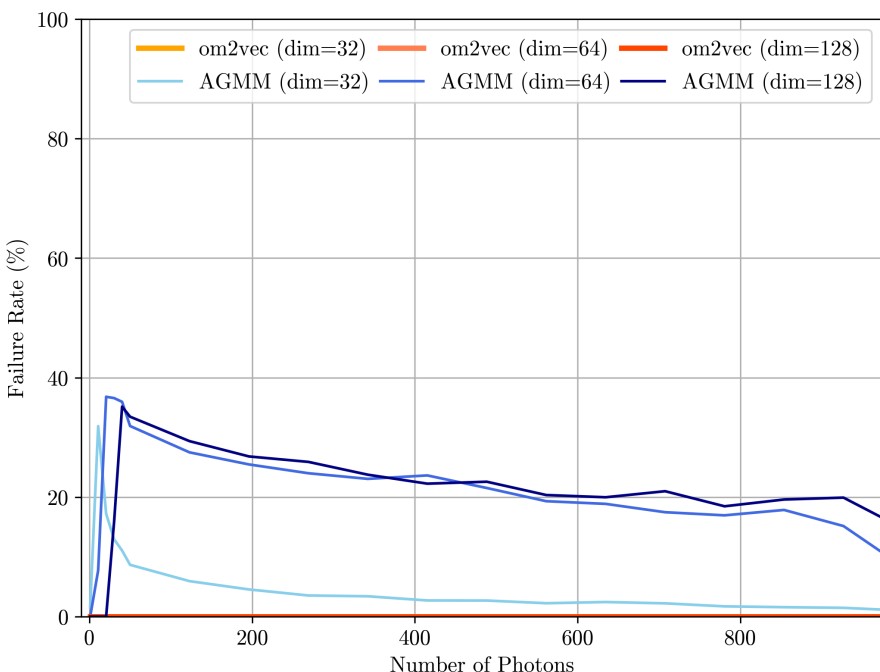

Figure 4: Failure percentage ($> 0.99$ JS distance) as a function of the number of photons on the input PATD. om2vec is always at 0% in this figure.

Notable characteristics of these curves include well-reconstructed distributions on the left, where the number of photons is low enough for the representations to "memorize" the distribution. This is followed by a sharp upward trend, peaking in a regime where there are too many photons for memorization, but insufficient statistics to generate a well-formed PATD. We then observe the JS distance decreasing as the number of photons increases, allowing for enough statistical information to effectively reconstruct the PATD. We also observe that increasing the latent dimension size for om2vec noticeably enhances the performance of representation learning, a trend not reflected in the AGMM method. Further insights into this phenomenon can be gained by examining the failure rate in Figure 4, where we define "failures" as PATDs reconstructed with a JS distance greater than 0.99. Notably, the VAE is never above this threshold for all PATDs. In contrast, a substantial percentage of PATDs from the AGMM method are mis-reconstructed due to optimization failures, with this failure rate increasing as more parameters and components are added.

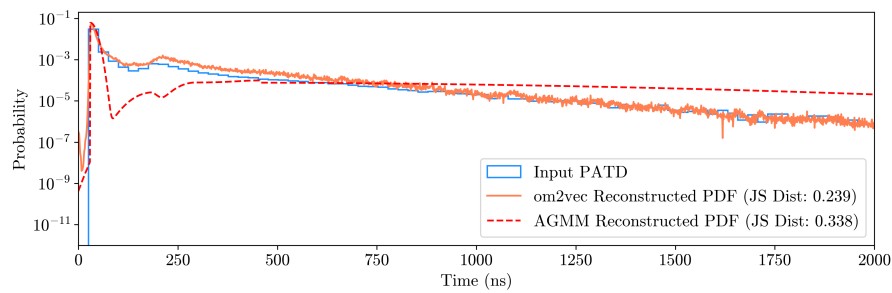

Figure 5: An example PATD of a "double-bang" signature, characterized by two distinct peaks. `om2vec` is able to reconstruct both peaks, while the AGMM is noticeably less accurate.

In order to give a more intuitive sense of the performance of `om2vec`, we examine individual PATDs and their reconstructed probability density functions. This is especially interesting for exotic cases such as the "double-bang" signature produced by $\nu_\tau$ interactions, as mentioned earlier. Figure 5 illustrates an example of this type of PATD, distinguished by its two distinct peaks. It is evident that `om2vec` reconstructs the bimodal structure of the timing distribution with greater precision than the AGMM method, despite the overall JS distance difference between the two methods being minor. As this is an extremely large PATD ($\sim$100,000 photons), this is likely due to the dominant statistics contained in the first peak. However, from a physics perspective, retaining information about both peaks is crucial for downstream tasks that attempt to isolate $\nu_\tau$ interactions, a smoking-gun indication of astrophysical neutrinos.

Table 1: Average JS distance and forward-pass FLOPs for the different `om2vec` models tested.

| MODEL | JSD | FLOPs |
|---|---|---|
| Fully-connected | 0.3545 | $2.899 \times 10^7$ |
| Transformers (1 block) | 0.2340 | $1.299 \times 10^9$ |
| **Transformers (3 blocks)** | **0.2177** | $\mathbf{1.882 \times 10^9}$ |

We further evaluated three different model architectures, all employing 64 latent parameters, to assess their impact on the average JS distance. As a baseline case, we trained a model using only fully-connected layers. The other two architectures incorporated transformers, as it is described in Section 2, with varying numbers of encoder and decoder blocks. A summary of the results is provided in Table 1. We note that the default architecture used in previous experiments features three encoder-decoder blocks and is highlighted in bold. Comparing the fully connected model to the transformer-based architectures, we observe a substantial improvement in reconstruction performance when incorporating transformer layers. However, this enhancement comes at a considerable computational cost, with the number of FLOPs required for a forward pass increasing by approximately two orders of magnitude. Furthermore, adding additional transformer encoder-decoder layers yields a slight further improvement in reconstruction performance.

## 5.2 RECONSTRUCTION WITH REPRESENTATIONS V.S. FULL INFORMATION

We now evaluate whether downstream physics analysis tasks suffer any performance loss when using the latent representations. One of the most critical downstream analysis tasks is angular reconstruction, i.e. predicting the direction of the incoming particle based on the photons recorded across the OMs. We test three different separately trained ML models for angular reconstruction on track-like events.

- `SSCNN (Full)`: a sparse submanifold convolutional neural network (SSCNN) that utilizes full timing information in a 4D CNN, as described in Yu et al. (2023).
- `SSCNN (om2vec)`: another 3D (reducing the timing dimension) SSCNN that leverages the latent representations from `om2vec`.

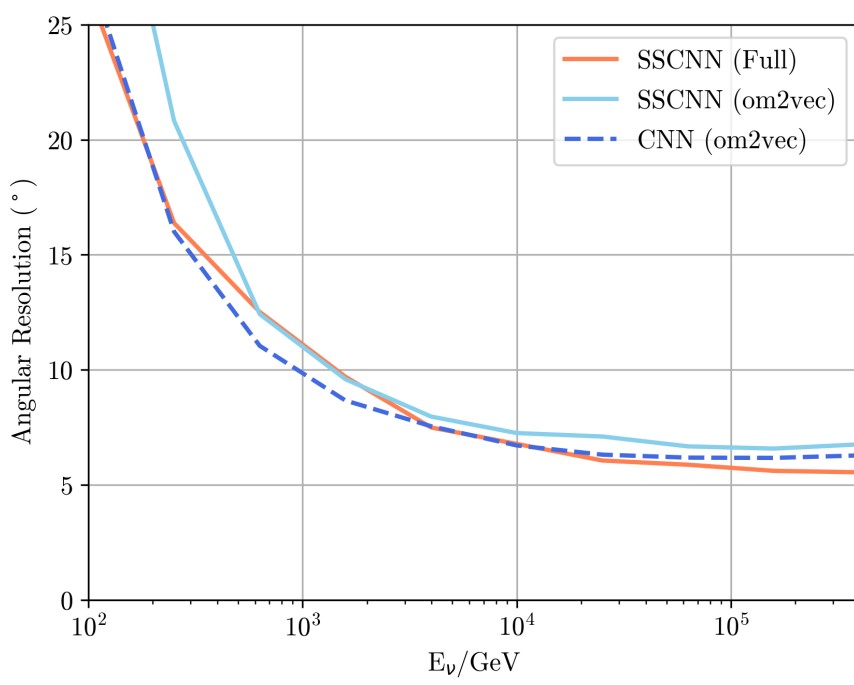

Figure 6: Angular resolution as a function of the true neutrino energy. The models trained using the latent representations from om2vec are able to perform just as well when compared to using the full timing information.

- CNN (om2vec): a traditional 2D ResNet CNN, where each neutrino event is encoded into a 2D image representation. These image representations are created by pixelizing the detector geometry along the strings and the OMs per string, then concatenating the latent representation and spatial position information at each pixel for the feature dimension.

In the methods employing om2vec, we first pre-process the events using the trained 64-parameter om2vec model, to generate latent representations for the subsequent SSCNN and CNN networks. In Figure 6, we show the neutrino angular resolution (the angular difference between the true and predicted neutrino direction), as a function of the true neutrino energy. We find that the SSCNN models using the full timing information and the latent representations perform similarly, though as expected, the full 4D model is able to achieve a slightly better resolution. Notably, SSCNN sacrifices some performance to remain computationally feasible on higher-dimensional sparse data (Graham & van der Maaten, 2017). Thus, we observe that the dense CNN approach with latent representations performs just as well or slightly better than the SSCNN with full-timing information, depending on the true neutrino energy. This CNN approach is made possible by the fixed-length latent representations. Importantly, we note that this new flexibility allows for the easy adaptation of more powerful models, such as vision transformers (Dosovitskiy et al., 2021), for angular reconstruction.

## 5.3 RUNTIME EFFICIENCY

In terms of forward pass encoding runtime, om2vec proves to be an efficient method, particularly when compared to classical approaches like the AGMM. Table 2 presents the average per-PATD runtime for various encoding methods alongside their latent dimensionalities. om2vec also significantly benefits from GPU acceleration, unlike the AGMM. Even when running on a CPU, om2vec is an order of magnitude faster. This is particularly important when running in resource-constrained environments such as IceCube's lab at the South Pole. With GPU acceleration, this speed advantage increases to two orders of magnitude. It is also worth noting that PATDs can be batched and processed in parallel on the GPU, which would further speedup the average runtime. Additionally,

Table 2: Average per-PATD runtime of encoding methods.

| METHOD | CPU RUNTIME (s) | GPU RUNTIME (s) |
|---|---|---|
| AGMM (dim=32) | 0.142 | - |
| AGMM (dim=64) | 0.557 | - |
| AGMM (dim=128) | 1.408 | - |
| om2vec (dim=32) | 0.0207 | 0.00184 |
| om2vec (dim=64) | 0.0204 | 0.00185 |
| om2vec (dim=128) | 0.0330 | 0.00193 |

om2vec scales well with increasing latent dimensionality, allowing users to expand the latent space with minimal runtime impact.

We also observe runtime improvements when running angular reconstruction algorithms on events using om2vec-represented PATDs instead of full-timing information. SSCNN (Full) took approximately 8.5s to process 20,000 events. In contrast, SSCNN (om2vec) took only 2.1s for the same number of events, representing a significant speedup. All GPU runtime tests were conducted on an NVIDIA A100 80GB.

## 6 CONCLUSIONS AND FUTURE DIRECTIONS

This work introduces om2vec, a novel approach utilizing transformer-based VAEs to efficiently represent neutrino telescope events. Our method addresses the significant challenges posed by the large size, high dimensionality, and variance of data in neutrino telescopes. By learning compact and descriptive latent representations, om2vec offers several advantages for downstream analysis tasks.

We have demonstrated that these learned latent representations not only preserve critical information from the original PATDs but also provide enhanced flexibility and substantial computational benefits. Our experiments show that models trained on these latent representations can achieve comparable performance to those using full-timing information in crucial tasks such as angular reconstruction. The enhanced efficiency and adaptability of om2vec encoding for neutrino telescope events paves the way for more sophisticated ML algorithms to be applied to neutrino telescope events. In particular, employing image-based algorithms in neutrino telescopes becomes straightforward by encoding the data using om2vec.

Another future direction could involve exploring data throughput rate reduction through latent representations. In existing active telescopes like IceCube, OM readouts are restricted by data throughput rate limits imposed by various physical and networking factors. Considering that high-energy events can result in tens of thousands of photons hits on a single OM, utilizing latent representations could decrease the throughput rate, enabling experiments such as IceCube to store higher-resolution timing information than is currently feasible.

By bridging the gap between the complex, high-dimensional data of neutrino telescopes and efficient ML techniques, om2vec represents an important step forward in our ability to sharpen our view of the neutrino sky. As our telescopes grow larger and data rates increase, our capability to efficiently process events will play an increasingly crucial role in pushing the boundaries of neutrino astronomy.

**Availability:** Source code, datasets, and pre-trained checkpoints for om2vec on Prometheus events is made available on GitHub. Additionally, as an immediate next step, om2vec is being integrated into GraphNeT (Søgaard et al., 2023), an open-source deep learning pipeline widely utilized by various collaborations in the neutrino telescope community.

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
