# OpenReview forum: "Learning Efficient Representations of Neutrino Telescope Events"
_ICLR.cc/2025/Conference — Submitted to ICLR 2025_

### Official Review · Reviewer_nXo1 · 2024-11-02

**Soundness:** 2
**Presentation:** 1
**Contribution:** 1
**Rating:** 1
**Confidence:** 4

**Summary:**

The paper titled "Learning Efficient Representations of Neutrino Telescope Events" introduces a novel approach called om2vec, which utilizes transformer-based variational autoencoders (VAEs) to effectively represent neutrino telescope events. The study addresses the challenges posed by high-dimensional, variable-length Photon arrival time distributions (PATDs) recorded by optical modules in neutrino telescopes, particularly focusing on the IceCube Neutrino Observatory.

**Strengths:**

- The use of a transformer-based variational autoencoder (VAE), called om2vec, represents an innovative approach for neutrino event data analysis, which has traditionally relied on more conventional statistical methods or simple summary statistics.
- The paper pushes the boundaries of machine learning applications within high-energy physics, specifically neutrino detection.
- By applying a VAE with transformer components to a unique scientific data source, the paper contributes to bridging techniques between disciplines, such as physics, machine learning, and data science. This could encourage further cross-disciplinary research and adaptation of machine learning models to complex scientific problems.

**Weaknesses:**

- The paper lacks a clear structure and does not adequately address related work. If this is indeed the first study applying deep learning techniques to the domain of neutrino telescopes, it is essential to include a dedicated **Related Works** section to provide context for this research.

- The figures in the paper are oversized. I recommend the authors resize them to a more standard dimension to enhance the overall presentation quality. The current size does not meet the standards expected for conference presentations.

- There are several typographical errors throughout the paper (e.g., lines 127, 484, etc.), which detract from its readability and should be addressed to improve clarity.

- The objective function is unclear, and the problem is not well-defined. The paper jumps directly to the results, with only a brief discussion of the classical $KL$ divergence. A significant improvement is needed in presenting a comprehensive **Proposed Methods** section that clearly defines the final objective function, rather than merely referring to it in the **Results section** (lines 228 to 230).

- Some statements in the paper are ambiguous or inaccurate. For example, the assertion in lines 223 to 232 that "the re-parameterization trick is utilized to construct the latent representation $z$, a vector of user-defined length referred to as the latent dimension. This technique guarantees that the latent space remains continuous and that similar representations within this space reconstruct to similar PATDs" is misleading and not entirely accurate.  However, the reparameterization trick separates the randomness of sampling (handled by $\epsilon$) from the parameters $\mu$ and $\sigma$, which allows to compute gradients with respect to these parameters. I recommend that the authors deepen their understanding of this concept from this paper [1].

I would be willing to consider increasing my rating, but only if these issues are adequately addressed. As it stands, the current version of the paper is not ready for publication.

**Refrences:**

[1] Kingma, Diederik P., and Max Welling. "An introduction to variational autoencoders." Foundations and Trends® in Machine Learning 12.4 (2019): 307-392

**Questions:**

The paper is somewhat limited as it presents results solely based on training and testing with simulated events, which may not accurately reflect real-world measurement data. Given that the approach uses a VAE-based transformer, it may perform better with simulated data that follows known distributions. Do you have access to any existing real-world datasets? If so, I would appreciate your feedback on this aspect.

---

> ### Author Response · Authors · 2024-11-26
>
> Hello, we would like to thank the reviewer for their suggestions! Below, we address their questions and comments:
> - The introduction has been rearranged and several sentences have been changed/added. This includes the addition of a dedicated related works subsection directly following the introduction.
> - All figures in the paper have been reduced in size by 20-25%.
> - To address the typographical errors, we have re-written the sentences regarding the source code/dataset availability on GitHub.
> - The training details section has been revamped, including the addition of a new section called “Proposed Methods”, which more explicitly defines the objective function as the reviewer suggests. This new section includes a mathematical definition of the reconstruction loss as well as some additional discussion about the KL divergence.
> - The ambiguous/misleading sentences about the VAEs have been re-written and adjusted: “The re-parameterization trick is then utilized to construct the latent representation z while maintaining proper gradient flow to these parameters. A key property of VAEs over regular autoencoders is their continuous latent space, meaning that similar representations within this space correspond to similar reconstructed PATDs.”
> - Regarding real-world data and testing, this is certainly something we have considered, and there are plans to apply om2vec to real-world experimental data. However, the experimental data is restricted and not publicly accessible. Therefore, it is common practice in the field to use simulation data, which typically provides a highly accurate representation of real-world performance.

---

### Official Review · Reviewer_sTJ4 · 2024-11-02

**Soundness:** 2
**Presentation:** 2
**Contribution:** 2
**Rating:** 3
**Confidence:** 4

**Summary:**

This develops a variational autoencoder to create a generative model for data produced by neutrino telescopes. The architecture is based on transformers, and results in a flexible representation and improved computation.

**Strengths:**

The application is certainly interesting and compelling. I also like the rationale of the work. There's a clear scientific motivation for these problems.

**Weaknesses:**

Several aspects. First, this is an ML focused conference so I would have appreciated greater details on the encoder and decoder without having to dig through the source code. Why transformers as opposed to a simpler architecture? Is there some kind transformation of the features that would allow for an MLP. Even if not, I would appreciate these as baselines as opposed to a traditional statistical model when comparing performance.

Also having worked with these a lot, I'm willing to bet that there was a substantial amount of tweaking required for learning rate and architecture parameters. If not, I'm certain performance can be improved dramatically by taking these steps. Another example, the runtime isn't really compelling to me. This is a feed-forward network, clearly it's going to be quicker than the alternatives. Should be supplementary, which would make more space for the fitting details I discussed.

Overall, this seems written for a scientific audience rather than an ML audience. I very much appreciate the application and clear motivation so I hope it's resubmitted. It just seems like some of the details we find interesting were glossed over and need to be improved for this to be accepted.

**Questions:**

Not at the moment, will see other reviewers' comments.

---

> ### Author Response · Authors · 2024-11-26
>
> Hello, we would like to thank the reviewer for their suggestions! We have conducted an additional study comparing a simpler feed-forward network with the transformer models and have found there are significant gains in the reconstruction ability. These results are summarized in the new Table 1.
>
> For hyperparameter tuning, we performed basic manual adjustments to identify the optimal parameters. The learning rates and the beta regularization factor were particularly sensitive. We also believe there is significant potential for improvement through more advanced parameter-tuning techniques. However, the primary aim of this paper was to introduce a novel model to a new field and demonstrate its application within that context. We felt that the runtime analysis was an important part of this demonstration, as it is relevant for deploying these models in real-world applications. Additionally, the runtime is significant because state-of-the-art technique normally cannot run with full PATD information due to computational restraints. Our technique is unique in this sense as it provides an approximation for this, and bypassing this computational restraint.

---

> > ### Comment · Reviewer_sTJ4 · 2024-11-26
> > **Needs a better methods section**
> >
> > Based on the other reviewer comments I believe that this paper is not strong enough for acceptance. The proposed methods section is a total of two paragraphs with no description of why this architecture was chosen over a vanilla VAE. It seems to rely on the very interesting application to compensate for minimal methodological contributions. This would be a perfectly acceptable decision if this were submitted to a paper in the field, but here we expect more on the mathematics/architecture side. I concur with nXo1 that a substantial revamp of the proposed methods would be required for an ML venue.

---

### Official Review · Reviewer_nfb2 · 2024-11-04

**Soundness:** 4
**Presentation:** 4
**Contribution:** 4
**Rating:** 8
**Confidence:** 4

**Summary:**

This paper presents om2vec, a novel approach leveraging transformer-based variational autoencoders (VAEs) to create compact, descriptive latent representations of photon arrival time distributions (PATDs) from neutrino telescope events. The proposed model is designed to handle the high-dimensional, variable-length data typical of neutrino observatories like IceCube. om2vec aims to outperform conventional approaches, such as asymmetric Gaussian mixture models (AGMMs), by improving reconstruction accuracy, runtime efficiency, and reliability while being less dependent on hyperparameters. The paper details the architecture, training, and testing with simulated datasets, comparing the method’s performance with traditional AGMMs and exploring its utility for downstream tasks like angular reconstruction.

**Strengths:**

- Originality: Applying transformer-based VAEs to neutrino event data is novel and demonstrates a creative extension of ML techniques to physical sciences.
- Quality: Comprehensive evaluation of the model against AGMMs, showing significant improvements in reconstruction accuracy, computational efficiency, and robustness.
- Clarity: The architectural details, data processing steps, and experimental methods are described with clarity, making the paper accessible to readers familiar with ML and neutrino physics.
- Significance: The ability to improve data processing and enable better downstream analyses has substantial implications for neutrino research and potentially for other high-dimensional physics datasets.

**Weaknesses:**

- Generalizability: While the results are promising, it would be helpful to see a more extensive discussion on how the method might generalize across different types of neutrino observatories or non-simulated real-world data.
- Comparison Baseline: Although om2vec is compared with AGMMs, additional comparisons with other potential ML approaches (e.g., deep CNNs or LSTMs) for PATD representation might strengthen the case for its use.
- Hyperparameter Sensitivity: While the model claims reduced dependence on hyperparameters, an exploration of performance variability with different encoder/decoder block configurations or latent dimension sizes would provide deeper insights into its stability.

**Questions:**

1. How does the model’s performance vary with different encoder/decoder block architectures or deeper networks?
2. Can the approach be adapted or extended to handle data from other types of particle physics experiments with different signal characteristics?
3. Have real-world data tests been considered, and if so, what were the challenges and results?
4. Is there potential for this method to contribute to real-time data processing in neutrino observatories under field conditions?

---

> ### Author Response · Authors · 2024-11-26
>
> Hello, we would like to thank the reviewer for their suggestions! Below, we address their questions and comments:
> - In regards to the comparison baseline, based on other reviewer comments, we have added a comparison to using just a fully-connected network for PATD representation, the new results can be found in Table 1. We have also added comparisons with different encoder/decoder blocks in the same table, in regards to hyperparameter sensitivity and the first question.
> - Yes, we believe that this methodology should be generalizable to other types of experiments with 1D waveform-like data. We specifically tried to keep this project as detector-agnostic as possible for this purpose.
> - Regarding real-world data and testing, this is certainly something we have considered, and there are plans to apply om2vec to real-world experimental data. However, the experimental data is restricted and not publicly accessible. Therefore, it is common practice in the field to use simulation data, which typically provides a highly accurate representation of real-world performance.
> - Real-time data processing is also something we have considered. The main threshold here is that the processing needs to be fast enough to handle event data rates at neutrino telescopes, which typically records thousands of events every second. Based on the runtime analysis demonstrated in the paper, there is definitely potential for this to be run at real-time, with some fine-tuning.

---

### Official Review · Reviewer_mVDH · 2024-11-04

**Soundness:** 2
**Presentation:** 1
**Contribution:** 2
**Rating:** 3
**Confidence:** 4

**Summary:**

This article presents an approach to learning representations of neutrino events by leveraging a transformer-based variational autoencoder. The model is trained to capture the photon arrival time distribution, and the learned representations are evaluated using the Jensen-Shannon divergence to assess reconstruction quality. Furthermore, the authors explore the applicability of these representations in a downstream task – angular reconstruction.

**Strengths:**

The application of machine learning techniques in scientific research is a vital and rapidly evolving field. We are delighted to see submissions in this area and encourage researchers to share their relevant work.

**Weaknesses:**

This article requires significant improvements in its writing and technical accuracy. Numerous technical details are either unclear, incorrect, or require further clarification (see Questions for specific concerns). As it stands, the article's technical clarity is compromised, which may lead to confusion and misinterpretation. A thorough revision is necessary to ensure the article's technical details are accurate, clear, and concise.

**Questions:**

* In Fig. 2, the “autoencoder” outputs some probabilities through the softmax activation. This is a confusing design. How is the reconstruction loss applied in this case?
* In section 4.2, the training methodology for the three models and the utilization of om2vec are unclear. Can you provide a more detailed explanation of the training process and how om2vec is incorporated?
* Are there any additional physics features that could be included in the time series data, beyond the current single feature of photon hits?
* In lines 179-180, the authors wrote “We opted for a learnable memory embedding for the transformer decoder layers, ensuring that the decoder portion of the architecture remains entirely independent of the encoder”. Please elaborate on the memory embedding block about its design.
* The model and training details in Table 1 are incomplete and unclear. Can you provide a more comprehensive description of the model architecture, including the number of encoder and decoder layers used?

---

> ### Author Response · Authors · 2024-11-26
>
> Hello, we would like to thank the reviewer for their suggestions! Below, we address their questions and comments:
> - We use the softmax activation as a normalization to ensure the output probabilities all sum to 1. We then use a negative log likelihood loss that compares the true and reconstructed PDFs, both normalized to 1. Some text has been added to clarify this: “After the final linear layer, the outputs are fed through the softmax function to obtain a properly normalized probability density.”
> - A more detailed discussion has been added in Section 4.2: “In the methods employing \texttt{om2vec}, we first pre-process the events using the trained 64-parameter \texttt{om2vec} model, to generate latent representations for the subsequent SSCNN and CNN networks.” We also added a sentence describing the binned Poisson likelihood treatment: “We interpret the final softmax activation's output as the probability of detecting a photon hit in each timing bin”, as well as a more in-depth discussion of the loss function in a new section “Proposed Methods”
> - We chose to only use photon hits as input features as it is a shared and universal feature that all neutrino telescope sensors have. However, on a detector-by-detector/experiment-by-experiment basis, one could think about including information relating to the specific sensor properties. For example, once could include the temperature and quantum efficiency of each OM, or, for water-based detectors, the local position of each OM (as OMs are not stationary in the water-based detectors). For the purposes of this study, we wanted to remain on detector-agnostic grounds.
> - On lines 179-180, we try to make it clear that we use a learnable parameter to feed in as the “memory” for the transformer decoder layer. However, the current sentence could be made more clear, so it has been changed to: “We use a simple vector of learnable parameters, the "memory embedding," which acts as the memory input for the transformer decoder layers.” We also removed the parts discussing the direct utilization of the decoder on the latent representations, which is a given for autoencoder type architectures.
> - Table 1 has been overhauled, as other reviewers have suggested, to include tests for different numbers of layers and architectures!

---

### Meta-Review · Area_Chair_9nSu · 2024-12-10

**Metareview:**

This is a very interesting submission, focused primarily on an application of representing neutrino events.  The submission is presented from a fairly informal level, with a very large amount of space devoted to the relatively non-technical Figure 1.  Reviewers were generally quite positive about the topic of the contribution, but nevertheless had significant reservations regarding its acceptance to ICLR in its current form.  Quoting from a review: "This article requires significant improvements in its writing and technical accuracy. Numerous technical details are either unclear, incorrect, or require further clarification...."

**Additional Comments On Reviewer Discussion:**

Authors provided a brief response to each reviewer.  Reviewer sTJ4 provided a succinct response summarizing the remaining concerns: that the methods section is just a couple of paragraphs long, and is missing information.  The submission aimed for a high level description of an interesting application of ML to physics.  This was appreciated, but the balance between accessibility and technical content seems to be misjudged, and a significant reworking of the presentation would be necessary for the submission to be appropriate for ICLR.

---

### Decision · Program_Chairs · 2025-01-22

Reject